# A Review of Fingerprint Sensors: Mechanism, Characteristics, and Applications

**DOI:** 10.3390/mi14061253

**Published:** 2023-06-14

**Authors:** Yirong Yu, Qiming Niu, Xuyang Li, Jianshe Xue, Weiguo Liu, Dabin Lin

**Affiliations:** 1School of Optoelectronic Engineering, Xi’an Technological University, Xi’an 710032, China; yirongyuu@icloud.com (Y.Y.); lixuyang@xatu.edu.cn (X.L.); wgliu@163.com (W.L.); 2BOE Display Technology Co., Ltd., Beijing 100176, China; xuejianshe@boe.com.cn

**Keywords:** fingerprint sensors, optical, piezoelectric, Internet of Things, ultrasonic

## Abstract

Identification technology based on biometrics is a branch of research that employs the unique individual traits of humans to authenticate identity, which is the most secure method of identification based on its exceptional high dependability and stability of human biometrics. Common biometric identifiers include fingerprints, irises, and facial sounds, among others. In the realm of biometric recognition, fingerprint recognition has gained success with its convenient operation and fast identif ication speed. Different fingerprint collecting techniques, which supply fingerprint information for fingerprint identification systems, have attracted a significant deal of interest in authentication technology regarding fingerprint identification systems. This work presents several fingerprint acquisition techniques, such as optical capacitive and ultrasonic, and analyzes acquisition types and structures. In addition, the pros and drawbacks of various sensor types, as well as the limits and benefits of optical, capacitive, and ultrasonic kinds, are discussed. It is the necessary stage for the application of the Internet of Things (IoT).

## 1. Introduction

The biometric traits of individuals include a wealth of information and are the basis for a variety of well-established biometric technologies. As a new information authentication technique, biometric identification based on human features may significantly improve the quality of life for individuals and foster economic growth. Recently, biometric identification technology has been utilized not only by government agencies to identify persons and protect public safety but also by businesses and social groups for a variety of commercial purposes. As a result of the development of biometric identification technology, biological traits provide major benefits, such as passwords, and human biometric characteristics for identity authentication have attracted considerable attention. Biometric identification mainly includes iris recognition [1,2,3], facial recognition [4], voice recognition [5], retina recognition [6], palm print recognition [7], vein recognition [8,9], fingerprint recognition [10,11,12], and so on. Biometric technology has vast applicability and business potential. Physiological and behavioral features are constant and distinct, and the likelihood of two individuals having identical physiological characteristics is quite remote.

The fingerprint is a pattern created by the ridged skin at the tip of a human finger [13]. The fingerprint is formed before birth, and its shape does not alter as the individual matures [14]. There are several methods for acquiring fingerprint pictures, which is the initial stage of fingerprint identification [15]. Many fingerprint acquisition methods exist: optical, capacitive, temperature [16], ultrasonic, and electromagnetic wave [17] fingerprint acquisition technology. After fingerprint photos are accepted, pre-processing is undertaken to simplify the extraction of fingerprint features, classifying fingerprints according to several characteristic points, then comparing them to those stored in the database to determine if they are the same.

Early fingerprint identification technology was mainly developed in cell phones. After Siemens demonstrated the feasibility of fingerprint identification technology in cell phones in 1998, companies such as Fuji, Motorola, and Apple began to produce fingerprint identification technology in cell phones, and so far, under-screen fingerprint technology is a major trend in cell phones. Fingerprint identification technology has developed particularly rapidly due to its great portability and high-cost performance, gradually becoming one of the more common biometric identification methods than others in life, e.g., attendance systems at work, smart door locks for family, identity determination in society, etc.

In recent years, fingerprint identification technology, from early ink identification to optical identification and capacitive identification, in particular the ultrasonic fingerprint identification system, has been gradually discovered by everyone. Ultrasonic fingerprint identification can reduce the error rate of fingerprint identification technology and improve the accuracy of fingerprint identification technology in the case of pollution identification. As the Figure 1 shows, the ultrasonic fingerprint sensor has high integration, which can effectively solve problems such as the size and accuracy of the fingerprint sensor. Fingerprint recognition technology is currently a convenient, highly reliable, and low-priced biometric technology with great potential for large-scale applications.

In this review, recent progress on fingerprints is summarized from research all over the world in more than 250 references, with 40% being published in the last five years. This work reviews the mechanism, characteristics, and industry application of fingerprint sensors. The first section outlines modern biometric technologies and describes the development and application of specific biometric technologies, which is helpful in understanding the background of fingerprint recognition technology. The second section discusses the mechanism of optical fingerprint sensors and details the structure of various types of optical fingerprint sensors, enumerating the characteristics and application fields of various types of sensors. The third section discusses the characteristics and application fields of various types of optical fingerprint sensors. The fourth section describes the structure and characteristics of fingerprint sensors based on self-capacitance and mutual-capacitance. The fifth section provides ultrasonic fingerprint identification technology based on various sensors. 

## 2. Biometric Recognition Mechanism

Recently, biometrics as a security technique has emerged [18]. Combining the physiological properties of the human body with computers, optics, and acoustics [19] led to the development of a range of biometric technologies, including iris, voice, fingerprint, face, etc., technology. Their comparison is shown in Table 1. Such features can be identified using advanced technologies to clarify personal identifying information [20]. Such physiological traits can provide stronger assurance for information security and lower the danger of data loss, leakage, etc. Moreover, biological features are a kind of characteristic information that accompany human beings, have high anti-counterfeiting performance, are difficult to fabricate or steal, are portable, and can be accessed at all times [21]. Nowadays, the research of biometric identification technology has made remarkable achievements in many fields [22]. For example, biometric identification technology plays a positive role in criminal identification and confirmation in the field of public security [23]. It can realize the encryption and decryption of relevant parts of the computer information system [24]. The use of biometric identification technology in the process of electronic commerce and monetary transactions can ensure the circulation of funds in social life [25]. Furthermore, it can provide remote medical monitoring [26] and other medical fields [27].

### 2.1. Iris Recognition

Iris recognition is a biometric system that identifies an individual based on the iris tissue in the human eye [28]. Iris fiber tissue is very intricate. The composition of its quantitative characteristic points may reach 266, which is an order of magnitude greater than comparable biometric technologies. To attain maturity, fiber tissue is created during the fetal stage, and the development process is highly unpredictable. Once generated, it cannot be altered, hence providing security, uniqueness, stability, unforgeability, and non-falsifiability [29]. According to the study, the iris can still be used for detection within 5–7 h to 21 days after human death [30]. The market for iris recognition technology is quite modest, although studies have demonstrated its viability on mobile phones [31] and in access control systems [32]. IOM (iris on the move) technology is a high-speed, easy, and high-throughput long-distance iris recognition security solution. It can reach a distance of 3 m to capture and analyze the iris photographs of several individuals.

Malgheet [33] has provided a summary of the many techniques for iris recognition research. Lei [2] proposed a model-agnostic meta-learning (MAML)-based several-shot learning method (new-shot learning) for iris recognition to solve the problem of a limited number of samples in deep learning for iris recognition technology; Sun [25] proposed an open-set iris recognition method based on deep learning that can effectively differentiate iris samples of location classes without impacting the known iris recognition capability. Wang [34] introduced a cross-spectral iris identification algorithm based on convolutional neural networks (CNNs) and supervised discrete hashing, which not only achieves superior performance than previously examined CNN [35] designs but also greatly reduces the template size.

### 2.2. Facial Recognition

The face is the most straightforward identifier of private information. Currently, facial recognition is prevalent, less expensive than other biometric technologies, does not require direct touch, and can be performed at a distance [3,36]. Multiples with a similar appearance, face occlusion, and facial photographs and videos might affect the accuracy of facial recognition. Fourati [37] distinguished faces in videos and images by image quality assessment and motion cues; Afaneh [38] provided matching decisions for multiples with a correct rate of more than 95% by constructing a biometric system; Kute [39] discovered local correlations in faces to achieve recognition and confirmation of obscured faces; Zeng [40] proposed a simultaneous occlusion invariant depth network (SOIDN) to simultaneously recognize and match unobscured and synthetically obscured faces and make full use of the relationship between the two; Zhang [41] proposed hierarchical feature fusion, which can be used in strong illumination and occlusion situations to improve recognition accuracy, especially to achieve good recognition results, and can be applied to smart cities; Madarkar [42] extracted the facial occlusion part to establish non-coherent samples, and improved the recognition of occluded faces through a non-coherent dictionary.

Expression recognition has a significant influence on the field of pattern recognition in facial recognition. Approximately 55% of human-to-human communication is transmitted by facial expressions, according to studies [43], and facial expressions are the best tool for identifying human emotions and intentions [44]. The identification of facial expressions is important to artificial intelligence and has enormous promise in psychological research, driver fatigue monitoring, interactive game creation, virtual reality [45], intelligent education [46], and medical fields [47]. After recognizing facial expressions, Wu comprehended the emotional content of images and generated image captions using the Face-Cap model [48]; Cha used surface electromyography (sEMG) around the eyes [49] (sEMG reference) to react to the user’s facial expressions [50], thereby performing expression recognition.

### 2.3. Finger Vein Recognition

Vein technology uses the veins beneath the epidermis, such as finger veins, hand veins, foot veins, and palm veins, for identification authentication [51]. Veins are fundamental properties that are exclusive to living beings [52] and are difficult to duplicate and forge [53]. Compared to fingerprint recognition, blood must be flowing in vivo to be detected; to a certain extent, the security degree is high. Additionally, the vein pattern is distinct even after several deliveries. Vein pictures of the hand are typically obtained under transmission illumination utilizing near-infrared (700–900 nm) [54], and the images can vary significantly according to the hand’s size, thickness, and placement angle. Compared to the same camera capture approach, Prommegger [55] utilized numerous cameras to acquire veins from different angles to increase the identification rate around the fingers. Despite its rarity, vein recognition offers a multitude of applications: Immanuel [56] used biometric authentication protection of finger veins for ATM network security. Su [57] integrated finger veins with ECG signals for personal identification, and the method was significantly better than separate systems in terms of recognition accuracy and security. Yang [58] encrypted medical data by vein recognition, and information was stored on a card to achieve convenience and privacy of information.

### 2.4. Voice Recognition

Voice recognition comprises speech and non-speech recognition, and voice recognition systems typically sample at a rate of 8000 Hz or more, with a frame size of 256 or 512 samples. It has several uses, including audio monitoring, sound event detection, and ambient sound recognition [59]. Hu [60] used pathological speech recognition to detect various vocal cord diseases while reducing laryngoscopy, which can be applied to medical facilities lacking laryngoscopy during telemedicine. Wijers [61] used a bio recorder to associate African lions with their vocals, find features, and locate and distinguish the vocal information transmitted to the lion’s roar. Nakamura [62] developed the ability to match and record an individual’s voice through accurate recognition of vocal features to improve the efficiency of the recordings. Beritelli [63] conducted identity verification based on the measurement of heartbeat frequency with an accuracy of up to 90%.

### 2.5. Fingerprint Recognition

Fingerprints are phenotypic genetic features that people are born with and form at the end of their fingers. Under normal circumstances, fingerprints vary, mainly in the detailed features of fingerprint ridges and valleys [64]. These detailed features are the key to the fingerprint recognition process. The fingerprint recognition process relies mainly on the bifurcation points and nodes in Figure 2. When the fingerprint is taken on the scanner surface, the ridge skin area is shown as black (or dark gray value), while the valley area and the background area are white (or light gray value).

Each individual has a distinct and stable fingerprint, recognizable by the irregular lines on the finger belly. Figure 2 depicts the different categories of fingerprint characteristics [65,66]: sweat holes, patterns of lines, early lines, scars, etc., on fingerprint lines. The earliest fingerprint recognition was collected with ink and paper, and the emergence of electronic computers spurred the development of fingerprint collection methods that enhanced the resolution and clarity of fingerprint images and were conducive to improving the processing efficiency of computers. Fingerprint sensors can employ the biometric code for a variety of purposes, including mobile phone unlocking [67], attendance systems [68], mobile payment [69], access control unlocking [70], ID cards [71], medical information [72], driver’s licenses, and so on. Its primary function is to identify personal data [73].

## 3. Optical Fingerprint Recognition

Optical fingerprint sensors, which use the principles of light refraction and reflection to produce pictures, have replaced ink for obtaining fingerprints since the invention of computers [74]. Charge-coupled devices are the central elements of optical scanning systems (CCD and CMOS). CMOS and CCD image sensor development began concurrently [75]. Due to the constraints of the process level at the time, CMOS resolution was low, there was a great deal of noise, light sensitivity was low [76], and the quality was poor; it also acheived little improvement. In contrast, CCD image sensors have dominated the market for two or three decades due to their broad effective light sensitivity area, uniform acquisition, low noise, and other advantages. Due to advancements in integrated circuit design technology and process levels over the past decade, CMOS’s previous problems have been substantially eliminated. Intra-pixel amplification, column-parallel structure, and random reading are incomparable to CCD [77]. CMOS sensors are smaller in size (to achieve the same image effect), consume less energy, and have improved integration and electronic-voltage conversion efficiencies [78,79,80,81,82]. Active CMOS image sensors (CIS) have developed rapidly in recent years [83,84,85] and have gradually become the primary choice for most imaging fields. Omni Vision, Micron, Panasonic, Toshiba, etc., have made significant breakthroughs in CIS pixels, making CIS development a later trend.

With the continuous development of optical fingerprint sensors, from single prism optical total reflection technology to optical thin film transistor (TFT) technology, and then to optical coherence tomography type technology, the resolution and clarity of the scanned image have been greatly enhanced, as has the success rate of fingerprint identification.

### 3.1. Single Prism Recognition Method

In 1971, optical fingerprint capture machines began obtaining pictures by scanning the fingerprint left on the surface of the finger, a process subject to interference. Various designs of optical fingerprint sensors emerged thereafter. Early optical fingerprint scanners typically required a separate light source and prism. Figure 3a is a typical total reflection principle, and this total reflection identification is affected by finger moisture and folds [86]. Figure 3b shows the imaging device placed beyond the critical angle, and only the light reflected on the ridge beyond the critical angle reaches the imaging device. Figure 3c employs the dispersion method inside the finger [87], wherein the light source penetrates inside the finger, scatters, and passes through the fingerprint layer to reach the transparent plate of the sensor and must have a refractive index near that of the human tissue. Figure 3d depicts a multispectral light source for detecting fingerprint pictures on the surface and within the fingerprint [88]. Because the transmission qualities of biological tissue depend on wavelength, various wavelengths penetrate to different depths, necessitating multiple light sources and two polarizers for this sensor [89]. In 2016, Baek created a fingerprint sensor based on the modification of the optical path [90] that enables the detection of wet fingerprints. Due to the sensor’s utilization of numerous optical components, its size is considerable. The necessity for minimal size has led to the development of linear scanning sensors [91], which suit both the need for mobile device integration and the demand for big fingerprint pictures. Most linear scanning sensors are sliding sensors, which introduce degrees of freedom as a result of changes in sliding speed and direction. Existing fingerprint sliding sensors can be categorized as capacitive or optical [92]. The optical type has a smaller size(4 × 0.9 mm^2^) and a greater resolution (1000(H) 625(V)), is made up of an optical binder with the same refractive index as the overlay glass, and has a single sliding direction [93].

### 3.2. Identification Method through TFT Technology

Thin-film transistors (TFTs) have been developed for over 40 years, replacing traditional semiconductor sensors with light-sensitive TFT panels. In 1961, Weimer created the first thin-film transistor (TFT) [94]. Using amorphous silicon (a-Si:H) as the active layer, Street created a TFT device in 1979, and it was discovered that amorphous thin-film transistors (a-Si TFT) might be employed as switching devices for active array liquid crystal displays (AMLCD) [95]. TFTs are produced by sputtering and chemical deposition methods on non-monocrystalline substrates such as glass or plastic wafers, and large-scale semiconductor integrated circuits (LSICs) are produced by treating the films. A standard TFT optical sensing device is comprised of a grid of pixels on glass and an operational amplifier (OPAMP) in an external readout integrated circuit (IC) [96]. This technique may be utilized to create large-area sensing arrays using optical fingerprint sensors. Table 2 lists the particular pairings for the four primary types of TFT technology: no crystalline silicon TFT, polycrystalline silicon TFT, organic TFT, and amorphous oxide TFT.

#### 3.2.1. Based on Amorphous Silicon TFT Technology

Hydrogenated amorphous silicon (a-Si:H) thin-film transistors (TFTs) are the predominant thin-film transistor technology used in flat panel displays today [97]. A-Si:H TFTs offer the benefits of low process temperature (<350 °C), good large-area uniformity, cheap cost, and low leakage current; they are the predominant technology for commercial LCD flat panel displays now. However, the electron field-effect mobility of a-Si:H TFTs is only 0.4–1.5 cm^2^/(Vs), and the hole field-effect mobility is considerably lower, which prevents their use in high-definition and current-driven displays. In addition, low mobility a-Si:H TFTs must be accomplished by extending the channel width to obtain adequate drive current. However, big-size a-Si:H TFT devices lower the display openness, resolution, and display brightness [98].

The absorption spectra and quantum efficiencies of a-Si:H and Si have been compared. In the range of 0.4–0.75 µm, the material absorbs lighter than the Si layer, but the Si layer absorbs more light at longer wavelengths (>0.75 m) than the a-Si:H film. In visible light, the a-Si:H-based diode generates a greater photocurrent than the Si-based diode, and a-Si:H pin diodes have a greater QE output than Si p+/n diodes over the majority of the visible spectrum [99].

Based on amorphous silicon TFT technology, optical fingerprint sensors may achieve slimness, compactness, and wide areas [100]. The PPS structure enables the creation of A-Si:H field-effect devices [101]. Thin optical touch (TOT), hidden optical touch (HOT), and hidden display (HUD) fingerprint sensors based on hydrogenated amorphous silicon are already available for mass production [99]. The sensor sizes range from tiny (4.0 mm × 8.0 mm) for smartphone user identification to the size of four fingers (3.2″ × 3.0″) for public safety inspection. Table 3 and Figure 4 provide a comparison of the three fingerprint sensors.

#### 3.2.2. Based on Polycrystalline Silicon TFT Technology

Low-temperature poly-Si technology refers to the process method for producing high-quality poly-Si films and thin-film transistors (TFTs) at temperatures below 600 °C [102]. On the same glass substrate, the pixel array and the driver circuit of a poly-Si TFT-LCD may be merged to accomplish the integration of the peripheral driver and display. Compared to a-Si:H TFT, LTPS TFT offers the benefits of high mobility (often two orders of magnitude greater) and big drive current, as well as reduced power consumption and a smaller device area; it also addresses the problem of high density to increase yield and decrease manufacturing costs [103]. Moreover, p-Si is resistant to light interference, and the leakage current does not rise under light circumstances, allowing the light-shielding layer to be omitted. It is possible to produce a high-definition display system on a panel (system on panel, SOP) that merges a display matrix and peripheral drivers, considerably enhancing the system’s dependability. LTPS TFTs have already replaced OLEDs in tiny commercial displays.

When amorphous silicon thin-film transistors shift to high capacity, high brightness, and high resolution, LTPS technology compensates for the smaller pixel size and shorter unit pixel charging time and also solves the problems of difficult high-density leads and integration of the display area with the surrounding driver circuit [104]. There are two ways to obtain poly-Si films: direct methods and indirect approaches. The direct methods are low-pressure chemical vapor deposition (LPCVD) [105] and catalytic chemical vapor deposition (cat-CVD) [106], and the indirect methods use amorphous silicon film recrystallization, whose crystallization methods primarily consist of solid-phase crystallization (SPC) [107], rapidly cooked annealing (RTA) [108], metal-induced lateral crystallization (MILC) [109], microwave crystallization [110], and laser crystallization (LC).

As the sensing material, a new photosensitive material is employed. The sensor’s metal/sensing material/ITO structure is incorporated into the LTPS TFT process [111,112,113]. Using an active pixel sensor circuit (APS), optical intracellular fingerprint sensor (iFP) technology that integrates display, touch, and FPS capabilities is created [114]. Meanwhile, amorphous silicon can be placed above polysilicon in a manner similar to Figure 5 to generate a vertical hybrid PIN photodiode (HPAS-PIN) [115]. The fingerprints that may be effectively acquired by this optical image sensor array are depicted in Figure 6a. However, this array cannot capture the fingerprint picture when a color filter is present. An LTPS image sensor (LIS) can be operated using an LTPS backplane-based active pixel sensor (APS) circuit [116] shown in Figure 6c. A 400PPI LTPS-TFT LCD with in-cell touch and in-cell screen fingerprint scanning capabilities has been designed, capable of fingerprint image capture even when the finger is at the height of 1.0 mm on the LIS array, with a fingerprint scanning area resolution of 256 × 256 and a sensor with enhanced sensitivity and interference resistance.

When the LCD is used as a light source, the LCD and its opaque backlight unit obstruct visible light, and there is an unavoidable air gap between the LCD and the backlight unit. An optical image sensor array can be built directly into the LCD with the backlight unit or an extra LED as the light source and the LIS array on a TFT substrate that eliminates the backlight as the light source. It is also feasible to position the IR CIS and IR light source under the backlight, with the IR light emitted by the IR light source passing through the LCD module and being received by the IR CIS upon fingerprint reflection.

Ye [117] proposed a collimating optical route based on a microlens array with an IR light source of 940 nm wavelength inserted beneath the cover’s lower plate. By regulating the exit angle and luminescence angle of the light source, the entire reflection of all oblique incident light on the top and lower surfaces of the CG can be achieved. As shown in Figure 6f,g,h the beam conveying the fingerprint information passes successively through each film layer before being caught by the sensor array beneath the display. Due to the size of the sensor, this lens unit must be constrained to a tiny region.

LCD in-finger detection versus OLED under-screen detection: OLED uses self-luminescence as its light source and employs micro-collimators to prevent the overlapping of light signals. Due to the under-screen design and self-luminous property, the depth-to-width ratio of the micro-collimation aperture can be chosen freely to achieve adequate collimation without allowing an excessive amount of stray light to enter the CIS.

Additionally, the researchers created optical fingerprint recognition [118] that can be integrated into LTPS image display technology beneath the OLED display and can cover the entire display’s dimensions. OLED uses its own self-illuminating light as its light source and employs micro-collimators to prevent overlapping light signals. The signal variation rate is improved by 50 percent by refining the process and circuit architecture. By improving the collimation technology, the signal-to-noise ratio is virtually doubled.

As depicted in the illustration, the collimation system consists of a collimation hole and a microlens. One approach involves installing the collimation system above the photo sensor, as depicted in Figure 6f,g, while another technique involves preparing the collimation system directly above the photo sensor, as depicted in Figure 6i,j,k, showing the SNR result of the collimation system. A collimation hole and a microlens are included in the collimation system. The light-receiving capabilities of the sensor can be enhanced by exposing and then immediately aligning the sensor’s collimation mechanism.

#### 3.2.3. Oxide TFT-Based Technology

Thin film transistors (MOS TFTS): Due to the benefits of good uniformity, high mobility, and strong process compatibility with a-Si TFT, the metal oxide TFT line drive circuit is often concerned with the following performance aspects, speed [121], power consumption [122], and reliability [123,124].

Traditional amorphous silicon (a-Si:H) TFT and low-temperature poly-Si (low-temperature poly-Si, LTPS) TFT have been proved challenging to employ to fulfill the need for flat panel displays with huge sizes and high resolutions. The semiconductor industry has reached an epoch in which material limitations restrict device size. Since 2003, transparent TFTs fabricated from amorphous oxide semiconductors have been a focus of worldwide research [125]. Compared to silicon materials, oxide semiconductor films exemplified by a-IGZO have superior low process temperature, long service life, high transmittance, large forbidden bandwidth (transparency), and high carrier mobility [126,127,128,129,130], and are widely applicable in liquid crystal displays, memory [131,132], and the Internet of Things [133]. Therefore, AOS TFTs are anticipated to replace a-Si:H TFTs as the primary devices for the subsequent generation of flat panel displays (flexible displays, transparent displays, etc.).

Qin [134] presented a near-infrared optical fingerprint sensor based on an array of passive pixel sensors (PPS) and oxide (IGZO) TFT technology. This optical sensor has an organic photodiode as its sensing element (OPD). This organic–inorganic hybrid thin-film photodetector is able to capture a crisp picture of a fingerprint. Additionally accomplished was a flexible fingerprint sensor based on a PI substrate. Due to the strong penetrability of near-infrared light, some of it can travel through the finger and land on the sensing element when the finger hits the sensor surface. In addition, the fingerprint’s ridges and troughs represent distinct optical routes. In the region of the ridge, a portion of the NIR light is reflected at the surface, while the remaining light falls on the detecting pixel. In the valley region, only a portion of the NIR light is transmitted into the air and subsequently reaches the detecting pixel. Through the ridge and valley sections of the fingerprint, different quantities of light are permitted to reach the surface of the sensor pixel accordingly. Therefore, fingerprint recognition is accomplished. Each pixel has a size of 50.8µm, equivalent to a resolution of 500 PPI.

#### 3.2.4. Organic TFT

The transition from TFT to OTFT is not easy. It is commonly accepted that Heilmeier [135] discovered the OTFT field effect phenomena on copper phthalocyanine thin film. In 1986, the first organic thin-film transistor was invented by Tsumura, who electrochemically synthesized polythiophene and employed it as the primary OTFT material; however, the final mobility was only 10^−5^ cm^2^/(Vs) [136]. On the other hand, because OTFT has good pliability, low-cost advantages, and the characteristics of organic materials themselves, different institutions at different times conduct in-depth research on thin film morphology and preparation techniques, and the application in the field of display devices will become increasingly widespread and significant.

Optical fingerprint biometrics based on organic photodiode (OPD) technology [119] may be embedded under the smartphone display to obtain complete coverage and capture up to four fingerprints [120]. As Figure 6o shows, the full-screen optical fingerprint module is readily integrated into mobile devices (attached to an OLED panel or with an air gap). It effectively prevents spoofing when applied to visible light (540 nm) and near-infrared light (850 nm and 940 nm). Figure 6l demonstrates that an Android application has been built for this purpose, allowing the user to register and validate the fingerprint with a matching time kept below 500 ms and a contrast-rich picture resolution of 5 Cy/mm.

Faced with the difficulty of full-screen TFT technology, Tai [137] opted for a gap-type TFT as the optical array system in Figure 6n. For fingerprint recognition applications, the high photocurrent of gap-type TFTs results in a big signal and rapid readout, which is a major benefit. Between the OLED screen and the gap-type TFT sensor array lies the collimator. Algorithms are used to the fingerprint picture for background removal and image enhancement. The fingerprint’s ridges and troughs are plainly apparent.

### 3.3. Identification Method by Optical Coherent Layer Scanning Technology

Modern civilization is becoming great interested in the development of fingerprint recognition systems with high security and resilience in the fingerprint identification process. The capacity of optical coherence tomography (OCT) to capture the depth information of the skin layers [16] has created a new study field for fingerprint recognition [138,139]. OCT was initially introduced to the field of fingerprint recognition for anti-spoofing [140], including automatic spoofing and live detection [141,142,143,144,145,146], internal fingerprint (i.e., subcutaneous fingerprint corresponding to live epidermis) reconstruction [147,148,149,150,151,152], and fingerprint identification/recognition [153,154,155,156]. Currently, confocal scanning OCT offers the greatest image depth and is much costly due to the inclusion of additional components such as lasers. Using a low-cost light source, such as a light-emitting diode (LED) or a heat source, it is feasible to perform OCT variant-full-field optical coherence tomography (FF-OCT). FF-OCT utilizes a camera with virtually instantaneous access to fingerprints and point-by-point image scanning to produce axial pictures, as opposed to typical confocal scanning OCT. The absence of pinholes in the detection path of confocal OCT and the shallow image depth are drawbacks of FF-OCT. However, FF-OCT has been shown to be effective in a variety of biological applications, including imaging of skin, brain tissue, the gastrointestinal wall, and the cornea.

Imaging methods based on optical coherence layers account for the effects of wetness, folds, and lack of contact. Using the varying transmittance of ridges and valleys, a red LED illuminates the nail side of the finger to form a picture of the fingerprint [87]. FF-OCT may also make use of an imaging Michelson interferometer and a silicon camera [157]. It comprises a small, lightweight LINOS opt mechanical system covered by a 30 cm × 30 cm × 1 cm Plexiglas panel. The sensor produces 1.7 cm × 1.7 cm pictures with a spatial sampling rate of 2116 dots per inch (dpi). The fingerprint is an exterior picture captured at 15 m. The picture displayed between 33–103 µm is the cuticle, and the sweat duct has white dots. The presented photos between 121–210 µm match the live epidermis. The 121–156 µm images are the ridges of the internal fingerprints, whilst the 191–210 µm depth images are the valleys, allowing fingerprint pictures to be produced [158]. Figure 7 shows a scanning picture of common fingerprints, moist fingerprints, and fingerprints based on an optical coherence layer.

Liu [159] suggested the optical coherence tomography (OCT)-based fingerprint identification system shown in Figure 7e–g. This device’s SD-OCT approach utilizes a light source centered at 840 nm; the light source light is generated by a super light emitting diode (SLED), and two identical telephoto lenses serve as a focusing lens and a scanning lens, respectively [160]. The gadget is designed for capturing by touch. During fingerprint capture, it is necessary to place the finger on a fixed cover glass. In contactless OCT, the use of a fixed cover glass eliminates the depth-dependent roll-off issue. The SD-OCT device’s axial and lateral resolutions are 8 µm and 12 μm, respectively. The apparatus records a 15 mm × 15 mm picture with an average scatter size of 24 μm. Each acquisition records a fingertip region of 1.8 mm × 15 mm × 15 mm with a spatial dimension of 500 × 1500 × 400 pixels.

## 4. Capacitive Fingerprint Recognition

As semiconductor [161] technology progresses and authentication devices in mobile devices such as smartphones and integrated circuit cards demand compact, low-cost packaging, capacitive fingerprint sensors are appearing. In principle, capacitive and inductive fingerprint sensors share a “flat” plate with hundreds of integrated semiconductor devices and a surface layer that is generally a few microns thick. The unevenness of the fingertip’s fingerprint, the actual distance between the bump and the bump touching the plate varies when the finger is placed on the capacitive sensor’s surface, results in different capacitance values [162,163]. The capacitance values are converted into current or voltage values, which are then converted into clerk data by an ADC. The greater the contrast of the fingerprint impression, the closer the fingertip is to the surface. The device completes the fingerprint-gathering process by averaging the gathered results.

The optical under-display fingerprint sensor exploits the difference in light reflected from finger ridges and valley areas to recognize fingerprints, but it has difficulty distinguishing dry fingers, which do not create regular and consistent contact with the sensor cover layer [86,164]. In terms of identification time, recognition of dry fingers, and manufacturing yields, ultrasonic under-display fingerprint sensors have the potential for development. In contrast, optical and ultrasonic screen sensors are exclusively compatible with OLED panels. On the screen, a reciprocal capacitive can function. Authenticity can be determined by detecting the impedance of fingerprints, which has different impedance frequencies [165] and different amounts of eddy currents due to the impedance turbo effect [166], as well as by a temperature sensor [167]. Capacitive fingerprint sensors still need great improvement in fingerprint detection.

The breadth of the ridge is between 100 m and 400 m, the depth of the valley is between 60 µm and 220 µm, and the width of the valley is between 75 µm and 200 µm [157,168]. When the ridge and valley are neighboring electrodes, the mutual capacitance differential can exceed 400 aF [169]. Through cost-sharing [92], charge transfer, feedback capacitive [170], and sample-and-hold schemes, capacitive sensors [171] may detect weak signals.

### 4.1. Fingerprint Sensor Based on Self-Capacitance

The capacitance between the measuring pin and the power source is monitored in the self-capacitance fingerprint sensor. When a finger is put on the sensor, its capacitance increases and the measured voltage increases and changes, enabling the detection of finger contact. In self-capacitive sensing, the unit cell typically consists of a sensor circuit based on Si technology [92,165,172], and the sensor’s capacitance may be directly and independently sensed and detected. Si-based fingerprint sensors can only be manufactured on opaque, brittle, and inflexible silicon wafers and thus can only be implemented in rigid devices, such as smartphone buttons. Self-capacitance is not compatible with multitouch functionality.

In the self-capacitance sensing approach, the cell typically comprises a Si-based sensing plate and a readout circuit [165,173]. High sensitivity may be achieved because the controller can be individually addressed, and the sensing electrodes can be directly and independently recognized. However, Si technology sensors can only be manufactured on opaque, brittle, and inflexible Si wafers, limiting their use to rigid devices.

Self-capacitance is not favorable to multi-touch and is also susceptible to ghosting issues. Oxide TFT is an advantageous option due to its medium mobility, high on/off ratio, cheap process cost, and suitability for transparent and/or flexible substrates [174,175,176,177,178]. To increase the safety level of self-capacitance frames per second (FPS), even at tiny scales, great resolution is necessary while keeping high sensitivity [179]. To guarantee the high resolution and sensitivity of FPS, oxide TFTs are therefore indispensable.

Oxide TFTs predominantly employ amorphous indium gallium zinc oxide (aIGZO) thin-film transistors (TFTs), which have an optical band gap of 3.05 eV compared to typical a-Si semiconductors (1.6 eV) and ensure up to 75 percent transparency. By eliminating the opaque fingerprint sensor array and accompanying control electronics, the display bezel’s size is drastically decreased. The results of the measurements reveal a distinct fingerprint picture, including the aperture. Depicted in Figure 8a, using only one a-IGZO TFT, a capacitive touch fingerprint sensor may be incorporated into a display, achieving a tiny sensor size, removing two bus lines, decreasing noise, and enhancing sensitivity while keeping the same amplification performance [180]. Based on a touch sensor, the capacitive fingerprint sensor (CFS) uses an amorphous indium gallium zinc oxide (aIGZO) thin film transistor (TFT) passive matrix [181]. Due to their high sensitivity, visual clarity, durability, and exceptional performance, such capacitive touch sensor (CTS) sensing systems (TSSs) are utilized in a variety of applications, including mobile devices, automotive, military, and industrial equipment [182,183,184,185,186,187]. This sensor minimizes the number of sensing lines in an active matrix, decreasing expenses. This sensor’s construction is depicted in Figure 8f: 2.8 mm cover glass, 0.1 mm top optically clear adhesive (OCA), 0.005 mm RX electrode, 0.21 mm top film, 0.05 mm bottom optically clear adhesive (OCA), 0.005 mm TX electrode, and 0.21 mm bottom film pixel; 500 PPI resolution and 50.8 µm × 50.8 µm unit pixel size. AFE IC compensates for CSTRAY in sensing mode, boosting the dynamic range to 40.78 dB with an SNR of 47.8 dB and 25.0 dB for CTS and CFS, respectively. Figure 8h demonstrates that readout integrated circuits (ROICs) with a pixel size of less than 44 µm × 44 µm per pixel and a resolution path of more than 500 DPI may be employed.

### 4.2. Fingerprint Sensor Based on Mutual Capacitance

In the smartphone market, all-screen devices with thin bezels and no physical buttons have become the norm. On smartphones, fingerprint unlocking positioned on the back or side of the device is significantly less successful than front-mounted unlocking. In the mutual capacitance fingerprint sensor, two electrodes are utilized: the transmitting electrode and the receiving electrode. The TX pin provides the digital voltage and measures the charge received on the RX pin; the charge received on the RX electrode is proportional to the mutual capacitance between the two electrodes, and when a finger is placed between the TX and RX electrodes, the mutual capacitance decreases, and the charge received on the RX electrode also decreases. By detecting the charge on the RX electrode, the touch/non-touch condition may be determined.

Mutual capacitance is easily applicable to fingerprint sensors with two electrodes and an insulator structure, one for driving and the other for sensing. Mutual capacitance makes multi-touch functionality possible [189,190,191]. Mutual capacitance sensors can operate on OLED (organic light emitting diode) and LCD (liquid crystal display) displays with detecting distances typically ≤ 300 µm [192,193]. On glass substrates or by adhering polymer films on window glass, these sensors can be manufactured [194]. Transparent capacitive on-screen fingerprint sensors created utilizing mass-producible materials and techniques are ideally suited for smartphones and other mobile devices and can also be utilized to create large-area fingerprint touch combination sensors and sensors for flexible/stretchable electronics.

Five varieties (thickness of these layers: 100 m) of transparent sensor films are available: PET, cellulose nanofiber (CNF) films, and CNF films embedded with BaTiO_3_. Three nanoparticles (Chinese fibers + BaTiO_3_, banknote iron content two nanoparticles: 1 wt%, average size: 25 nm) and CNF films are embedded with AgNF (CNF+AgNF, silver fluorine content: 1.2 wt%, average length of AgNF: 200 ± 20 µm, average diameter: 380 ± 35 nm) [195,196,197]. CNF films have advantages due to their high transparency and good mechanical flexibility and durability; however, the low dielectric constant of the original CNF films (k = 1.4–3.0) restricts their usage as overlays for fingerprint sensor arrays [198,199,200,201]. CNF films, including AgNF (CNF+AgNF), have a k value of 9.2 and a transmittance of at least 90%.

Transparent electrodes with high conductivity and optical transmittance (T) are required for the operation of capacitive fingerprint sensors in the high-frequency region in the transparent form. Thin layers of resistance (Rs) of standard transparent electrode materials prevent high-frequency signals from driving capacitive fingerprint sensors when mobile devices emit interference. To achieve great transparency with metals as electrodes, the width of the electrode lines must be restricted due to their opacity, according to the data in Table 4.

When the finger hits the contact surface, capacitive fingerprint sensors can also generate a potential difference through the piezoelectric capabilities of the material, in which the sensor is bent by stress. It is reported that zinc oxide nanoarrays, which gather the charge caused by the piezoelectric potential generated by each nano, can reconstitute the fingerprint’s three-dimensional deformation field [206]. For the sensor surface to be encased with a chemically inert UV-cross-linkable polymer [207], the encapsulation layer must have a thickness between 1.5 µm and 10 µm and possess excellent hydrophobicity and oleophobicity [208], as shown in Figure 9. Hsiung [209] proposed an 8 × 32 pressure sensor array, with a cell size of 65 µm × 65 µm, a sensitivity of 0.39 fF/MPa, a maximum capacitance variation of 16%, and a resolution of 390 DPI, with no further film processing needed. Sugiyama [210] presented a 32 × 32 silicon pressure sensor array with a sensor chip size of 10 mm × 10 mm and an array element spacing of 250 µm, capable of displaying stable 2D or 3D images.

## 5. Ultrasonic Fingerprint Recognition

The ultrasonic fingerprint sensor [211] is the most precise and accurate instrument for acquiring fingerprint images. There are two primary imaging techniques: pulse-echo imaging and impedance imaging. Ultrasonic fingerprint imaging is based on the reflection effect of ultrasound as it propagates across various impedance mediums [212]. When a finger is placed on a mobile phone’s touch panel, the pressure sensor detects the pressure and transmits an electrical pulse to activate the ultrasonic fingerprint sensor, which emits an electrical pulse wave. Due to the difference in acoustic impedance between human tissue and air, the echo amplitude of human tissue is larger than that of air; hence, the woven pattern can be determined by determining the echo amplitude at each point. The intensity of the sensor’s ultrasonic waves is comparable to medical diagnosis, which is safe for the human body [213]. Due to the fact that ultrasonic waves have a high penetrating power [214], they can be identified even when there is a small amount of dirt or moisture on the finger, and they can penetrate materials such as glass, aluminum, stainless steel, sapphire, and plastic, which increases the device’s applicability and success rate. It is also feasible to place the sensor into the gadget in order to increase its durability.

Ultrasound imaging relies on the detection of ultrasound pulse echoes, with hills and valleys providing distinct echo signals. It is also possible to study acoustic impedance approaches by measuring the ring drop of the pulse, which measures the damping of the contact region (i.e., the fingertip) [215]. Acoustic impedance-based fingerprint recognition relies on the direct touch method with air-filled valleys of 430 Rayl and ridges of about 1.5 MRayl of human tissue, and the difference in amplitude and phase of the impedance helps to differentiate between ridges and valleys, thereby generating a fingerprint image.

The primary component of the ultrasonic fingerprint sensor is the ultrasonic transducer, which captures the ultrasonic signal impulse response and completes the graphic reconstruction using the reflection and diffraction properties of ultrasonic waves. Ultrasonic fingerprint recognition technology is unaffected by surface clutter and can penetrate the dead skin layer to reflect the fingerprint pattern architecture in the dermis, capturing not only the fingerprints visible on the surface but also obtaining credible information from within the tissue.

The earliest ultrasonic fingerprint sensor is a type of ultrasonic probe detection (US5224174A) jointly proposed by Ultrasonic Scan and Niagara and is a probe ultrasonic detection system [216]. The probe emits ultrasonic energy and scans twice from two directions at right angles; after the finger reflects, the pulse receiver absorbs the reflected signal and translates it into frequency data, and a processing circuit creates the fingerprint image. With the introduction of micro electromechanical system (MEMS) technology, the MEMS ultrasonic transducer stage into CMOS has made ultrasonic fingerprint recognition applicable to smartphones [217]. Capacitive micromechanical ultrasonic transducers (CMUT) and piezoelectric micromechanical ultrasonic transducers are typical transducers (PMUT).

### 5.1. Fingerprint Sensor Based on Capacitive Ultrasonic Transducer

A capacitive micromechanical ultrasonic transducer (CMUT) has a simple design, a compact size, low noise, high sensitivity, high resolution, and excellent matching between silicon material and dielectric impedance [218,219,220]. CMUT can be used to manufacture conventional semiconductor processes due to its simpler structure and developed technology, and it entered the market before PMUT. Metal uppers are a standard component of CMUT’s fundamental structure. Metal upper electrode, vibrating film, edge support, vacuum cavity, insulating layer, bottom electrode, and base constitute the standard CMUT construction (North Central University thesis) [221]. The lower electrode is attached to the base, whereas the upper electrode is capable of vibrating with the membrane. In the transmitting mode, the alternating electric field between the upper and lower electrodes causes the vibrating membrane to bend and vibrate, thereby generating sound waves; in the receiving mode, the vibrating membrane vibrates under the action of external sound pressure, causing the electric capacity between the upper and lower electrodes to change, and the electric signal corresponding to the magnitude of the sound pressure can be obtained through it.

Figure 10a depicts a CMUT composed of several tiny vibration cells (cells) connected in parallel, and multiple microelements are connected in parallel in a certain configuration to form a CMUT in order to obtain a greater output sound pressure. It can be designed in a variety of shapes, including circular, square, hexagonal, etc., depending on the specific requirements.

The first capacitive micromachined ultrasonic transducers (CMUTs)-based fingerprint sensor could only collect 2D fingerprint images via mechanical scanning. Savoia [223] avoided the requirement for mechanical scanning and provided a linear array of probes for a fingerprint sensor. The transducer is comprised of 192 rectangular elements with 112 µm element spacing and a total aperture of 21.5 mm, which corresponds to the average width of a human fingertip. This aluminum-encased, probe-type fingerprint sensor is utilized in medical therapy. Grayscale intensity representation of obtained 3D objects and 2D display of flat sections are also viable. However, the resolution of the photographs it captures must be enhanced. Pulse-echo measurements are not possible due to the array’s restricted bandwidth. The interface between this huge sensor array and the signal processing circuits is too complicated to meet the size requirements. Kwak [224] utilized waveguide obstacle mapping techniques for fingerprint imaging. It is possible to obtain fingerprint images from hard materials such as glass, which can also be used for fingerprint imaging under glass. Adjusting the waveguide’s height and width, as well as the medium’s composition, can enhance the resolution of a fingerprint image.

The waveguide method provides a higher resolution than the direct contact approach for detecting phase shifts of 0.6 degrees at ridges and valleys at 2.4 MHz. The enhanced fingerprint protrusion resolution safeguards and stabilizes the CMUT impedance signal.

### 5.2. Manufacture of Capacitive Ultrasonic Transducer

CMUT fabrication usually uses a silicon wafer as the base, aluminum material as the metal electrode, SiO_2_ as the edge support, and the vibrating film can be made of silicon, SiN, etc. On the silicon wafer substrate, a thin layer of metal is precipitated as the lower electrode pole plate of the vibrating unit. The insulation layer plays a protective role during etching and fabrication and also ensures the insulation of electrical characteristics between the upper and lower electrodes to prevent short circuits due to accidental joining of the upper and lower electrodes during operation. On the upper surface of the vibration film, a thin layer of metal is deposited, which is the upper electrode pole plate of the sensor. The edge support acts as a support for the vibration film [225].

The leads of the array are connected to the flexible part of the PCB, and a preformed feeler is cast onto the custom acoustic and mechanical backing of the CMUT chip to fix the rigid part of the PCB to the die. The silicon substrate of the chip is removed using the HNA wet etching process. CMUT is based on the localized oxide of silicon (LOCOS) direct wafer bonding. It consists of a silicon top plate, a vacuum gap for silicon nitride, and silicon oxide at the bottom. The vibrating top plate consists of a single crystal of silicon with a radius of 24 µm and a thickness of 1.5 µm below the flat plate with a vacuum gap of 200 nm. Fixed with epoxy resin to solder the 64-channel CMUT to the chip carrier, and the lead bonding end is also fixed with epoxy resin to prevent dislodgement.

### 5.3. Fingerprint Sensor Based on Piezoelectric Ultrasonic Transducer

CMUT is based on a capacitive electrode separated by a submicron vacuum gap, whereas PMUT is composed of a solid piezoelectric capacitor [226], which simplifies the production process and enhances mechanical stability. As depicted in Figure 10b, the PMUT consists of a piezoelectric membrane, upper and lower electrodes, and a membrane that vibrates. In the emission mode, a certain voltage is supplied between the upper and lower electrodes of the piezoelectric membrane, and the pressure generated by the membrane’s inverse piezoelectric effect forces the membrane structure to bend, causing the vibrating membrane to deform. When the alternating voltage is applied, the membrane vibrates and emits sound pressure, thereby converting electrical energy into sound energy. In the receiving mode, the vibrating membrane deforms due to the external acoustic pressure and causes the piezoelectric membrane to deform, thereby generating a corresponding charge due to the piezoelectric effect and enabling the conversion of acoustic energy to electrical energy and the reception of acoustic signals via the receiving circuit [227].

As illustrated in Figure 10c, the early arrays of AlN-based PMUTs for ultrasonic fingerprint sensors were only 24 × 8 [222]. Each PMUT is directly connected to a dedicated CMOS receiver amplifier, avoiding electrical parasitic and eliminating the need to pass via silicon vias. Fluoride FC-70 is utilized as the coupling layer between the PMUT and the finger, and a 100 µm PVC cover seals the fluoride and preserves the sensor [228]. To preserve the scratch resistance of the surface and reduce the impact of scratches and deformation on the sensor surface on the image, a protective layer can be added to the material surface [229]. Typically, a 1µm thick coating of Al_2_O_3_, a material with known hardness and scratch resistance [230], can be selected. It can be placed on plastic and glass, among other substrates. Thin enough layers of this material have little effect on the transmission of sound. In 2016, Horsley [231] proposed a monolithic 110 × 56-cell ultrasonic pulse-echo fingerprint sensor based on piezoelectric micromachined ultrasonic transducers (PMUTs) that were directly linked to a CMOS readout ASIC [232]. The array’s fill factor is 51.7%, which is three times greater than the 24 × 8 array. The ultrasonic fingerprint sensor captures not just surface-visible fingerprints but also information from within the tissue, demonstrating three-dimensional information with significant security enhancement [233]. Compared to the 110 × 56 array, the 65 × 42 array has a larger pitch to decrease crosstalk. The sensor is capable of multi-channel TX beamforming, which increases the signal-to-noise ratio by 7 dB and permits a little improvement in lateral picture resolution. With a five-column TX beam, peak pressure, RX voltage, and image contrast are all improved by a factor of 1.5. In the unsupported zone, the array’s amplitude is −17 dB, compared to −2.7 dB for 110 × 56. The sensor surface is initially covered with a 215µm-thick polydimethylsiloxane layer (PDMS). It is possible to create a 4.6 mm × 3.2 mm fingerprint image [234].

Typically, the piezoelectric coefficient of lead zirconate titanate (PZT) is two orders of magnitude greater than that of aluminum nitride (AlN). The PZT-based fingerprint sensor (50 × 50 PMUT array) employs a single pixel and a mechanical scanning mode of operation, resulting in a large size, poor frame rate, and high cost [235]. In Table 5, the various arrays are compared.

### 5.4. Fingerprint Sensors Based on Other Sensors

Typically, ultrasonic waves are produced by a transducer that converts electrical or other sources of energy into acoustic energy. Employing the inverse piezoelectric effect, the majority of ultrasonic transducers generate sound waves from an alternating electric field using piezoelectric materials. The same transducer can also receive ultrasonic waves through the direct piezoelectric effect. For the conversion of acoustic energy to electrical energy, piezoelectric nanomaterials are composed of lead zirconate titanate (PZT) and polyvinylidene fluoride (PVDF) nanofibers. The g_33_ value of PZT nanofibers (0.079 Vm/N) is significantly greater than that of bulk PZT (0.025 Vm/N) and PZT microfibers (0.059 Vm/N), resulting in an increased sensitivity for acoustic detection. Due to their high electrical coefficients and energy conversion efficiency, nanostructures based on PVDF are also commonly used as piezoelectric active materials. The combination of nanotechnology and ultrasonic technologies can alter standard ultrasound procedures. PZT-5 piezoelectric active material and P(VDF-TrFE) film were used to create ultrasonic fingerprint sensors by the Peng team [236,237]. The PZT-5H-based sensor effectively captured electronic images of the surface fingerprint pattern and the inner vascular simulation channel by simulating skin with two layers of polydimethylsiloxane (PDMS) and simulating blood vessels with bovine blood in the inner vascular simulation channel. The performance of the various sensors is compared in Table 6, where it can be seen that the PZT-based sensor structure has an order of magnitude higher emission sensitivity than the 110 × 56 PMUT-based sensor and is two orders of magnitude higher than the CMUT-based sensor and that the PZT-5 based sensor has the highest loop sensitivity of all designs. P(VDF-TrFE) films, unlike piezoelectric ceramics (e.g., PZT), are CMOS-compatible and can be directly integrated with supporting electronics [238]. Schmitt [239] suggested a new fingerprint sensor comprised of 1–3 piezoelectric ceramics that transfer the acoustic impedance of fingerprint patterns to the electrical impedance of the sensor element. The cost-effective sensor substrate delivers a resolution of up to 50 µm over a 25 × 25 mm^2^ area. It offers better electromechanical coupling, broader bandwidth, and lower acoustic impedance than piezoelectric ceramics, the traditional principal material (PZT).

Qualcomm released the ultrasonic fingerprint sensor that is currently prevalent in cell phones; the sensor is laminated directly on the OLED display to allow quick single-touch authentication with high speed, superior security, and low battery consumption. It occupies little space between the battery and the display, as its thickness is less than 150 µm. Strohmann [240] proposed the notion of contact gestures utilizing the Qualcomm fingerprint sensor system, and in addition to its authentication function, the same sensor can be used to recognize finger contact gestures and movement directions, such as light/heavy touch and finger navigation. Xu [241] presents a large-area (20 × 30 mm^2^), multifunctional, highly integrated ultrasonic sensor beneath a thick (>1 mm) mobile display made of organic light-emitting diodes (OLEDs) and protective cover glass (CG). Dual fingerprint verification enhances the sensor’s dependability and security by enabling simultaneous and rapid finger scanning to identify finger contact pressure levels and non-electromagnetic hard stylus position and tracking [242].

### 5.5. Manufactur of Piezoelectric Ultrasonic Transducer

Common PMUT diaphragm shapes include round, square, rectangular, shell, cylinder, and dome; piezoelectric membrane materials are shown in Table 7, including PZT [243], ZnO [244,245], AlN [246], PMN-PT [247,248,249,250], and PVDF [251], among others. The following table illustrates a comparison of various piezoelectric materials. Vibrating membrane materials are frequently silicon, silicon dioxide [252,253], silicon nitride, polysilicon, etc.; boundary conditions are approximate solid support, approximate simple support, approximate free vibration, etc.; vibration modes are double stacked bending vibration modes, multi-layer stacked bending vibration modes, thickness vibration modes, etc.; operating frequencies can range from tens of kHz to gigahertz, from the development of a single PMUT to the development of an array of PMUTs.

PMUT is generally based on silicon or SOI wafer-bonded piezoelectric materials. AlN is generally carried out by magnetron sputtering. AlN is often patterned by wet etching, using TMAH (C₄H_13_NO, tetramethylammonium hydroxide) solution as the wet etchant. AlN can also be patterned by dry etching, and the etching gas system can be BC_l3_ (boron trichloride)/C_l2_ (chlorine)/Ar (argon). Compared to wet etching, the steepness of dry etching is very high, and the surface roughness after etching is better.

In contrast, the PZT films fabricated by the magnetron sputtering method still have better film density and consistency at 6″ and 8″ sizes. Magnetron sputter-grown PZT films have better orientation growth characteristics, and despite their polycrystalline orientation, their internal domain orientation is highly uniform, such that the film-forming PZT can no longer be subjected to post-polarization. Of course, the problem with magnetron-sputtered PZT films is the higher cost, and the control of the growth environment and process conditions is much more complicated than that of magnetron-sputtered AlN [254].

After generating y piezoelectric material, it is necessary to pattern the piezoelectric material and the upper electrode. AlN is often patterned by wet etching, using TMAH (C₄H_13_NO, tetramethylammonium hydroxide) solution as the wet etching agent. AlN can also be patterned by dry etching, and the etching gas system can be BC_l3_ (boron trichloride)/C_l2_. Of course, the problem of magnetron sputtering PZT films is the high cost, and the control of the growth environment and process conditions is much more complicated than magnetron sputtering AlN. The wet corrosion of PZT is complicated, and the corrosion environment is mainly various acidic solutions. The dry etching of PZT, ICP (inductive coupled plasma), and RIE (reactive ion etching) are mainly mixed solutions. Wet etching can also be carried out. The choice of etching gases is also relatively large. SF_6_ (sulfur hexafluoride)/Ar, CF_4_ (carbon tetrafluoride)/Ar, and CHF_3_ (trifluoromethane)/Ar can be used for dry etching of PZT [255].

The ultrasonic fingerprint sensor is a current identification method with high accuracy. The advantages and disadvantages of different fingerprint identification principles are compared in Table 8. It can be seen from the table that ultrasonic fingerprint identification technology is a major development trend of fingerprint identification technology in the future. Wet or unclean fingerprints can be read by ultrasonic fingerprint sensors, and the ultrasonic type of flexible piezoelectric materials can be used in a variety of wearable sensors [256].

## 6. Conclusions

This work focuses on the current development status of fingerprint sensors, mainly describes three types of technologies currently available on the market, presents the structure and evolution of three distinct fingerprint sensor principles, and introduces the production methods for ultrasonic fingerprints.

Optical fingerprint sensors are mature and use in fingerprint sensors with large sizes, such as attendance systems, etc. The capacitive fingerprint sensor is further integrated on the basis of the optical type, which reduces the size to a certain extent and can be applied in portable devices such as cell phones. Recently, further research has been conducted on ultrasonic fingerprint recognition technology in order to reduce the error rate of fingerprint recognition and, at the same time, improve the integration of fingerprint recognition systems. For ultrasonic fingerprint recognition devices, ultrasonic transducers are mainly used to send and receive ultrasonic signals, and there are already ultrasonic transducers made of PZT, PVDF, and AlN. AlN was earlier used in piezoelectric ultrasonic transducers, PZT has cost advantages, and PVDF is mainly for flexible devices. For various applications of fingerprint transducers, different materials of piezoelectric materials can be selected.

Fingerprint recognition technology still has several shortcomings and directions for improvement:The accuracy of the algorithm is not sufficient to prevent recognition errors due to the proximity of fingerprints between relatives and needs to be improved;The fingerprint information left when touching an object is easily accessible, and the security is poor. Therefore, the detection of location authenticity must be enhanced to prevent the harmful effects of fingerprint theft;With the emergence of wearable devices such as mobile fingerprint unlock bracelets and car fingerprint locks, the integration of fingerprint recognition technology into flexible wearable devices has become a major challenge, which will drive the development of small, ultra-thin fingerprint capture chips;Fingerprint capture is easily affected by posture and angle, and the problem of finger pressure can be solved using a contactless fingerprint sensor.

Fingerprint recognition is a convenient and widely applicable biometric technology. A variety of different capture methods are committed to transferring fingerprint recognition devices from high-end devices to low-cost, miniaturized devices. In particular, fingerprint recognition technology can also be greatly applied to information security and medical testing of wearable devices. At present, ultrasonic fingerprint recognition technology is still a challenge to be widely used in life as one of the fingerprint recognition methods with better recognition effects. However, there is still lots of work to be carried out to achieve robust, interoperable, secure, privacy-preserving, and user-friendly systems.

## Figures and Tables

**Figure 1 micromachines-14-01253-f001:**
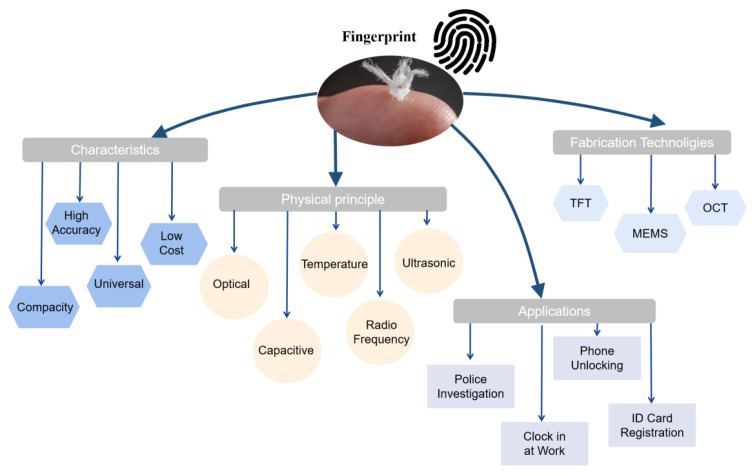
An illustration showing a summary of characteristics, physical principles, and technologies for the fabrication of fingerprint sensors in different applications. TFT (thin film transistor), MEMS (micro electromechanical system), and OCT (optical coherence tomography).

**Figure 2 micromachines-14-01253-f002:**
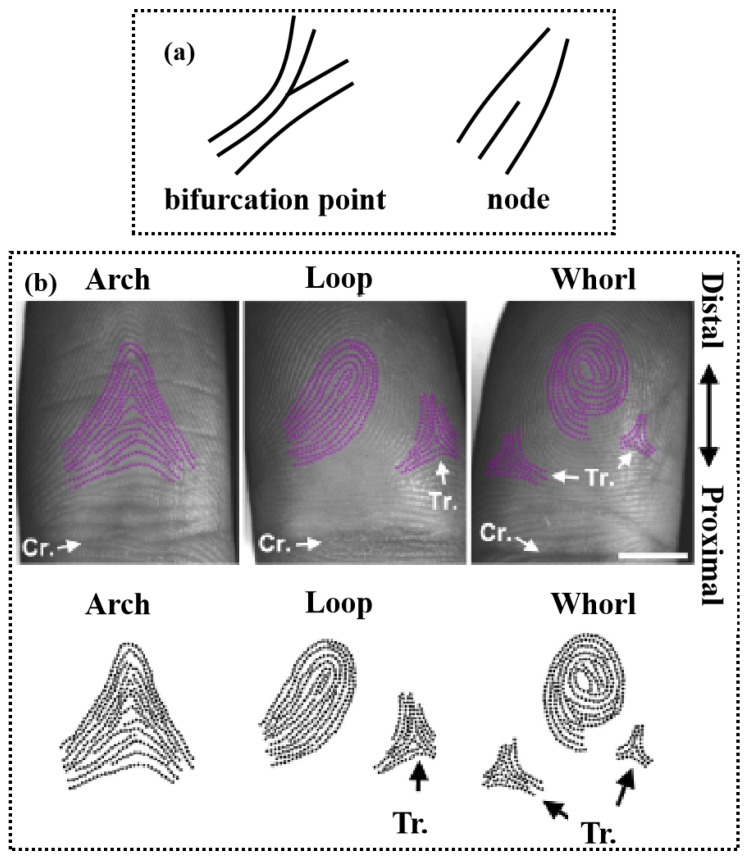
(**a**) Basic fingerprint feature points: bifurcation and node. (**b**) Arch, loop, and whorl patterns on adult fingertips. Tr., triradius; Cr., crease [13].

**Figure 3 micromachines-14-01253-f003:**
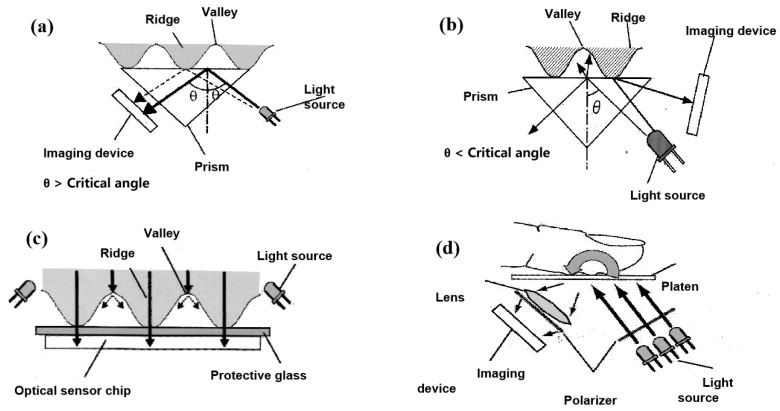
Many optical fingerprint sensor designs. (**a**) The principle of the whole internal reflection technique; (**b**) the principle of the light-path separation method; (**c**) the principle of the in-finger light dispersion method; (**d**) the principle of the multispectral method [89].

**Figure 4 micromachines-14-01253-f004:**
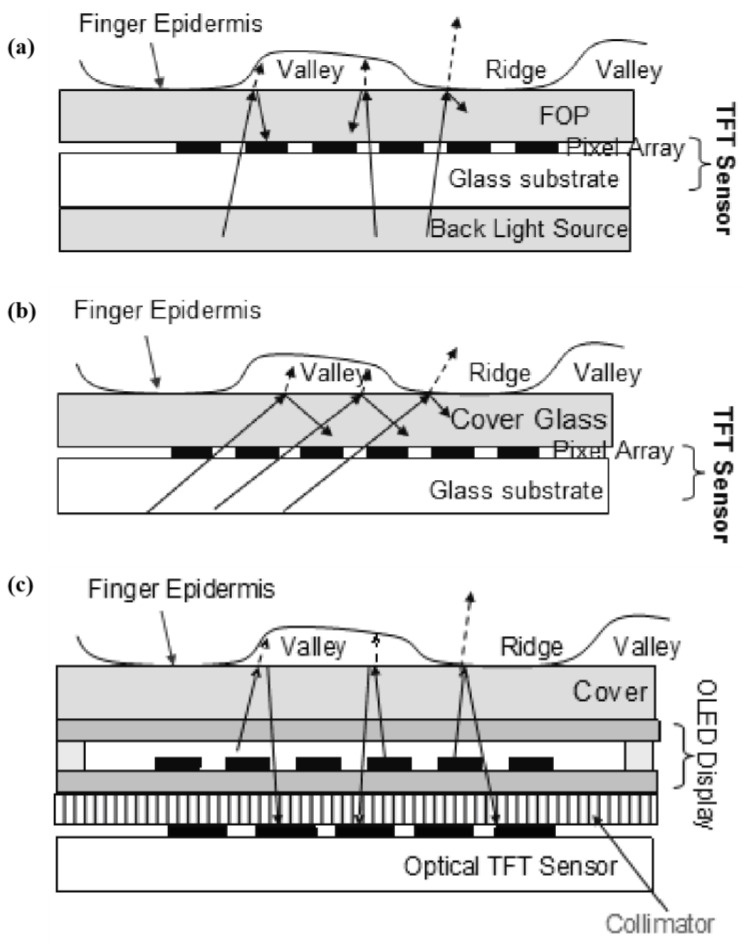
(**a**) the structure of TOT, (**b**) the structure of HOT, and (**c**) the structure of HUD [99].

**Figure 5 micromachines-14-01253-f005:**
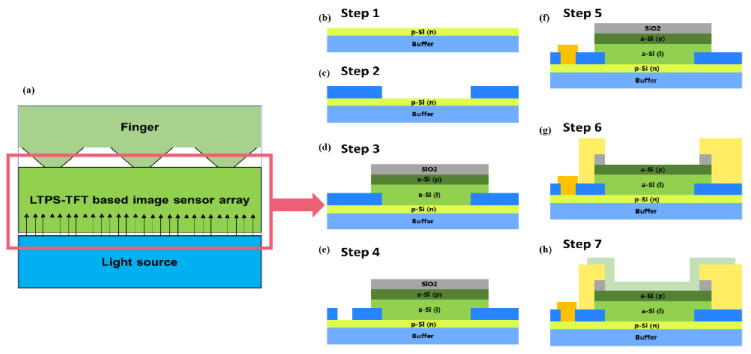
(**a**) Structure of optical image sensor array and fingerprint image captured by it. (**b**) Deposit a 300 nm oxide silicon buffer layer and a 50 nm intrinsic a-Si layer sequentially, then transform a-Si to p-Si by using Excimer-Laser Annealing. After that, phosphorus ions are doped into the p-Si to form an N+ p-Si layer. (**c**) Deposit a 230 nm insulating layer and form a 20 × 10 µm first via hole pattern by photolithography and dry etching. (**d**) A 600 nm intrinsic a-Si and 50 nm oxide silicon is deposited to cover the first via hole and insulating layer, followed by boron ions implanted to the surface of the intrinsic a-Si layer, turning part of the upper intrinsic a-Si layer into P+ a-Si layer. As ion concentration distribution is Gaussian distribution, no clear boundary between P+ a-Si and intrinsic a-Si exists. In this step, intrinsic a-Si in the first via contacts the N+ p-Si, together with the upper P+ a-Si layer, to form a hybrid p-Si and a-Si PIN photo-diode. After that, define the a-Si to a 25 × 15µm island by photolithography and dry etching. (**e**) Form a second via hole of insulating layer next to the first via hole using the same photolithography and dry etching. (**f**) Deposit a metal layer on the second via hole to serve as the cathode contactor of the photo-diode. (**g**) Coat an organic flat layer and form a third via a hole above the first hole. In this step, the oxide silicon is removed to expose the surface of P+ a-Si at the same time. (**h**) Deposit a transparent conductor layer ITO to serve as anode contactor of the photo-diode [115].

**Figure 6 micromachines-14-01253-f006:**
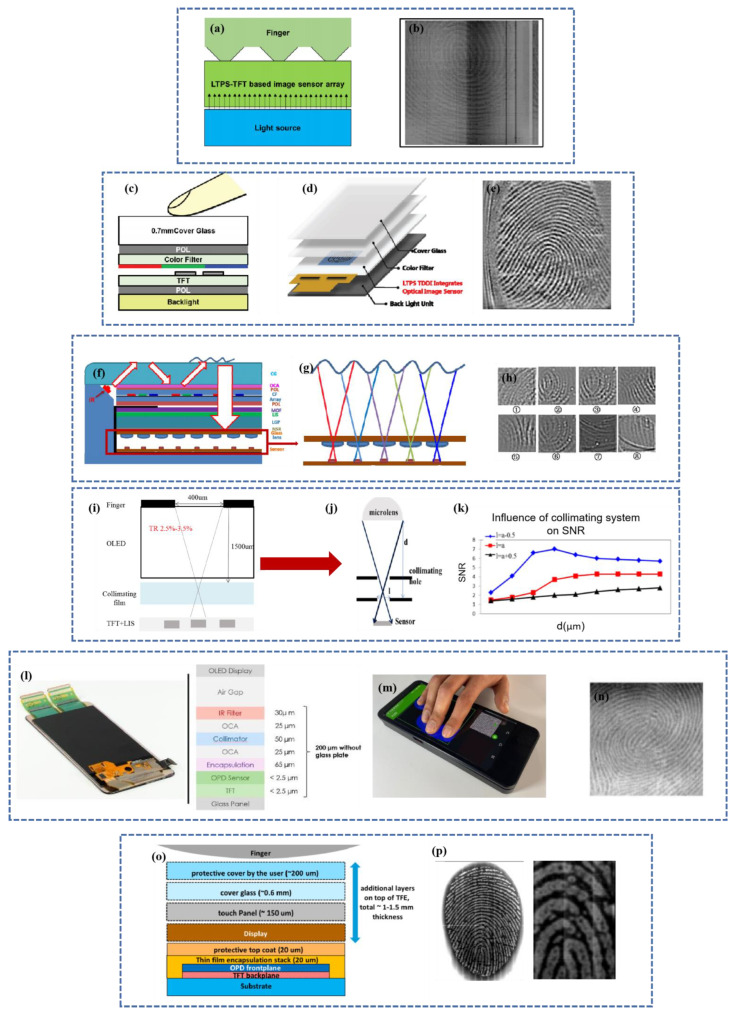
(**a**) Structure of optical image sensor array and (**b**) fingerprint image captured by it [115]; (**c**) cross-structure of in-cell FPS. (**d**) Structure of LCD with in-cell FPS; (**e**) the fingerprint image [116]. (**f**) Field of view diagram of the micro-lens array, (**g**) a collimating optical path based on microlens array, (**h**) the fingerprint image captured by it [117], (**i**) Stack structure of the fingerprint under OLED, and (**j**) diagram of a collimating system on the photo sensor. (**k**) Influence of collimating system on SNR [118], and (**l**) fingerprint on display (FoD Module). Assembly stack of the FoD module, (**m**) construction of organic image sensor, (**n**) the fingerprint capture by it [119], (**o**) structure of fingerprint sensor, and (**p**) the fingerprint capture by it [120].

**Figure 7 micromachines-14-01253-f007:**
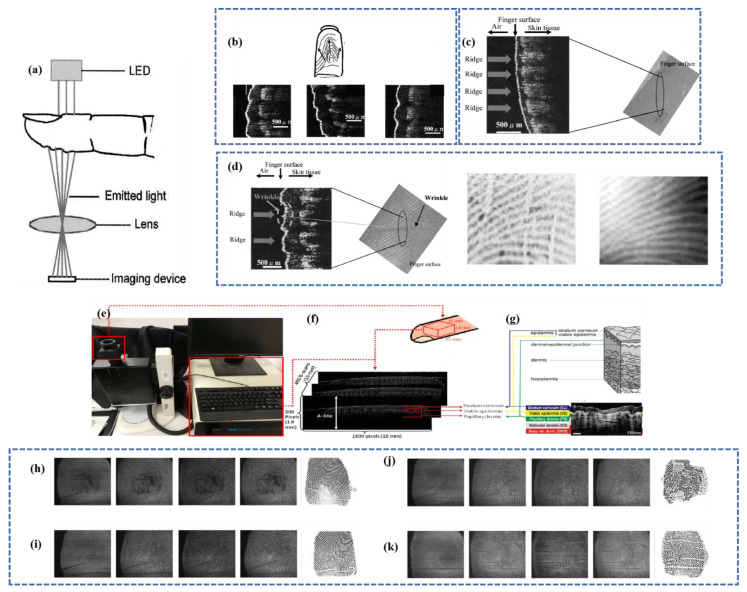
(**a**) Construction of a novel fingerprint sensor using scattered transmission light. (**b**) Normal fingerprint and the OCT images. (**c**) Fingerprint images of a wrinkled finger and the OCT image. (**d**) Almost flat finger image and the optical coherence tomography (OTC) image [89]. (**e**) Construction of equipment, (**f**) structure of subsurface fingerprints, (**g**) different layers of subsurface fingerprint, and (**g**–**k**) different fingerprint images. The images from left to right correspond to the cornea, internal, papillary, fusion, and traditional 2D fingerprints [159].

**Figure 8 micromachines-14-01253-f008:**
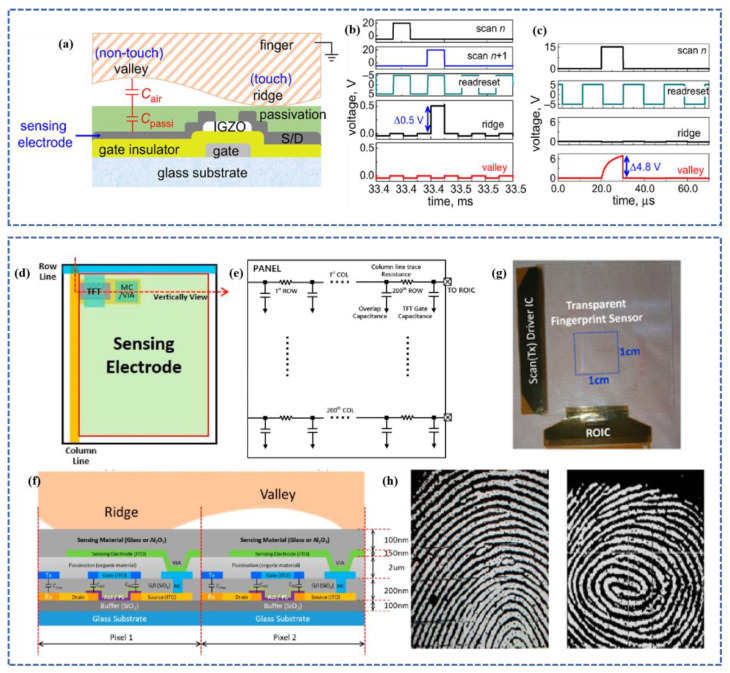
(**a**) Principle of capacitive touch-fingerprint sensor. (**b**) Simulation results for the capacitive sensor for touch-fingerprint circuits [180]. (**c**) Simulation results for the capacitive sensor for the proposed circuit, (**d**) top view of the proposed sensor pixel, (**e**) the circuit of the sensor, (**f**) cross-sectional view of the proposed sensor pixel, (**g**) the assembled module of the transparent fingerprint sensing system, and (**h**) the image of the fingerprint [188].

**Figure 9 micromachines-14-01253-f009:**
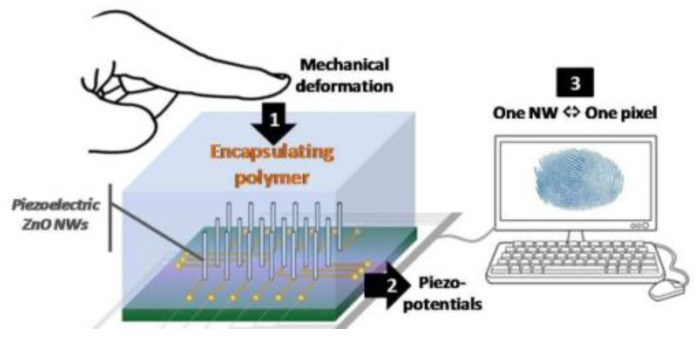
Schematic representation of polymer-encapsulated multi-NWs pressure-based fingerprint sensor [208].

**Figure 10 micromachines-14-01253-f010:**
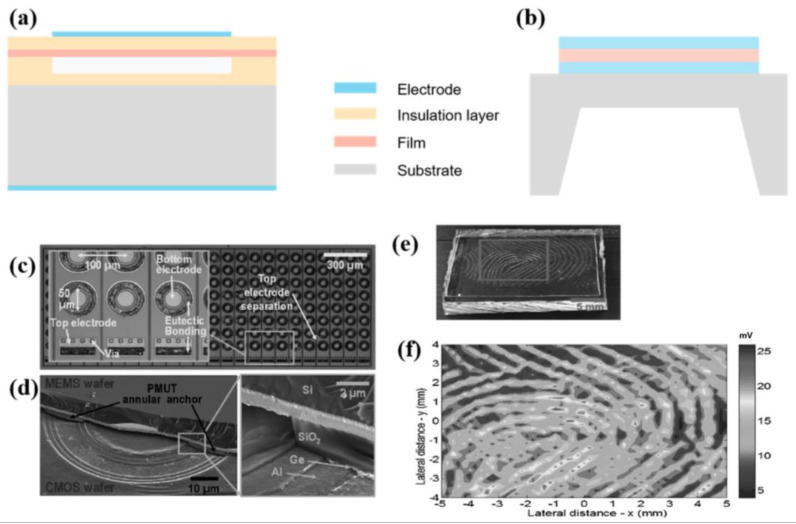
(**a**) The structure of CMUT; (**b**) the structure of PMUT. (**c**) Optical images of the 24  ×  8 PMUT array after de-bonding to remove the CMOS wafer, (**d**) cross-sectional SEM images of a single PMUT after partial de-bonding to remove the MEMS wafer, and (**e**,**f**) 2D pulse-echo ultrasonic image of the PDMS fingerprint phantom [222].

**Table 1 micromachines-14-01253-t001:** Comparison of different biometrics.

Comparative Aspects	Method	Market	Speed/Person	False Rejection Rate/%	Advantage	Disadvantage
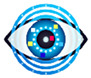	Iris	optical	about 7%	1–25 s	About 10	Not easy to age and wear	Difficult collection
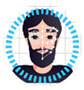	Facial	optical	about 18%	≤5 s	<0.2	Non-contact	Affected by the light, posture, and facial expression
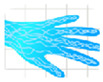	Finger vein	optical, capacitive, ultrasonic	about 3%	1–10 s	5	A wide range of identification	Large-size
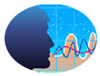	Voice	magnetoelectric, capacitive.	about 5%	1–3 s	About 10	Non-contact	Affected by the volume, speed, and sound quality of the sound
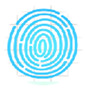	Fingerprint	optical, capacitive, ultrasonic	about 58%	≤1 s	About 5	Small equipment	Marks on the screen

**Table 2 micromachines-14-01253-t002:** Different types of TFT.

Semiconductor Materials	Process Temperature/°C	Migration Rate/cm^2^∙V^−1^∙s^−1^	Number of Lithography	Capacity
Amorphous Silicon	<350	0.1–1	4–6	high
Poly silicon	<700	10–400	5–11	low
Organics	<150	<2	-	low
Amorphous Oxide	<350	1–100	4–7	high

**Table 3 micromachines-14-01253-t003:** The structure of TOT, HOT, and HUD.

Sensor Type	TOT	HOT	HUD
**Light source**	460 nm LED	Wavelength invisible LED	OLED
**Light source position**	Back of the TFT sensor	Under the TFT glass substrate	-
**Sensing area**	FAP10(0.5″ × 0.65″)~FAP60(3.2″ × 3.0″)	10 mm × 14 mm	12 mm × 20 mm 12 mm × 40 mm 40 mm × 51 mm
**Bonding method**	Optical adhesives	Optically Clear Resin (OCR) Optically Clear Adhesive (OCA)	-
**Fiber optic board**	√	-	-
**Collimator**	-	-	√
**Sensor Top**	Fiber optical plate	A glass plate	OLED

**Table 4 micromachines-14-01253-t004:** Fingerprint sensor based on mutual capacitance.

**Area/mm^2^**	10 × 10	10 × 10	6 × 6	-	6.4 × 6.4
**Channel**	200 × 200	64 × 128	72 × 72	192 × 256	80 × 80
**Light Transmittance (%)**	94	94	-	79.90	89.05
**Electrode Material**	Indium tin oxide	Indium tin oxide	Metal Mesh	Indium tin oxide	Hybrid nanostructures
**Electrode Shape**	Diamond	Diamond	Half Diamond	-	-
**Capacitance/Voltage Difference**	50 fF (Ridge and Valley)	210 fF (Contact and non-contact)	-	4.2 ± 0.07 fF (Ridge and Valley)	-
**Resolution**	500 DPI	300–363 DPI	322 DPI	-	318 CPI
**Characteristics**	-	0.3–1 mm Cover glass	-	Acquisition of dual fingerprints	Pressure and temperature sensors
**Ref.**	[202]	[203]	[204]	[205]	[167]

**Table 5 micromachines-14-01253-t005:** Structure of ultrasonic fingerprint.

**Arrays**	24 × 8	110 × 56	65 × 42	50 × 50
**Top Electrode**	200 nm Mo	Al	24 µm Al	200 nmPt
**Piezoelectric Layer**	800 nm AlN	1 µm AlN	1 µm AlN	1 µm PZT
**Bottom Electrode**	200 nm Al	Mo	Mo	200 nm Pt
**Elastic Layer**	6 µm Si	2 µm Si	1.7 µm Si	10 µm Si
**Substrate**	SiO_2_	SOI	SiO_2_	600 nm Al
**Protective Coating**	Al_2_O_3_	PDMS	PDMS	-
**Filling Factor**	17%	51.70%	-	-
**Pixel**	-	591 × 438 DPI	376 × 318 DPI	-
**Readout Time (Individual\Array)**	-/24 µs	24 µs/2.64 ms	24 µs/1.56 ms	-
**Imaging Area**	2.3 × 0.7 mm^2^	4.6 × 3.2 mm^2^	4.6 × 3.2 mm^2^	-
**Resonance Frequency**	22 MHz	14 MHZ	20 MHz	25.02 MHz
**Array Spacing**	100 µm	43 × 58 µm	100 µm	50 × 100 µm
**Pulse Excitation**	28 V	24 V	24 V	-
**Number of Cycles**	2	3	-	-

**Table 6 micromachines-14-01253-t006:** Performance comparison of different piezoelectric materials.

Active Layer	PZT-5H	PZT-5	PVDF-TrFE	1–3 Piezoelectric Ceramics	110 × 56PMUT
**Resolution/DPI**	500	500 × 500	500 × 500	500	591 × 438
**Imaging range**	Fingerprints and finger vessels	Fingerprints	Fingerprints	Fingerprints	Fingerprints
**Bandwidth/%**	73.40	72.40	52.88	125	37
**Loop sensitivity/dB**	−52.84	−52.69	−60	−52.79	−78.06
**Sensor aperture size**	500 × 500 µm^2^	1 × 1 mm^2^	5 × 5 mm^2^	-	-
**Center Frequency/MHz**	21.2	20.7	39.85	16.1	14
**Conversion** **Sensitivity/kPa·V^−1^**	25.6	25.8	-	25.7	-
**Reception sensitivity** **/µ** **mV(kPa)^−1^**	89.1	89.9	-	89.2	-
**Impedance**	106	98.8	-	105	99.8
**Ref.**	[236]	[237]	[238]	[239]	[222]

**Table 7 micromachines-14-01253-t007:** Piezoelectric membrane materials.

Material Properties	AlN	ZnO	PZT	PVDF
**Electromechanical coupling coefficient/K^2^**	3.1–8	1.5–1.7	2–3.5	10–14
**Dielectric constant**	8.5–10	9.2	80–400	9–13
**Density/g·cm^−3^**	3.3	5.61	7.8	1.17–1.79
**Modulus of elasticity**	300–350	110–140	61	840
**Hardness/GPa**	15	4–5	7–18	-
**Coefficient of thermal expansion/ × 10^−6^ k**	5.2	4	1.75	-
**Piezoelectric constants/pC·N^−1^**	4.5–6.4	12	38–289	18.32

**Table 8 micromachines-14-01253-t008:** Compare that with the different types of fingerprint sensors.

Method	Core Technology	DPI	Cost	Default	Application
optical	Total reflection	low	low	Affected by the ambient light and the finger surface debris	roll machine, phone
capacitive	Intensive capacitance arrays	>500	low	Affected by the human body’s charge	phone
temperature	Micro heating element	300–500	middle	The temperature difference feature is weakened after multiple contacts	Carlock electronic key
ultrasonic	Ultrasonic imaging technique	>500	high	Prolonged exposure is harmful to the human body	phone
radio frequency	Radiofrequency imaging technology	high	high	Low recognition of some finger modes	True Print scanner

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
