# Peer review of "A Review of Fingerprint Sensors: Mechanism, Characteristics, and Applications"

_micromachines, 2023, doi:10.3390/mi14061253_

Round 1
Reviewer 1 Report
The primary problem of this submitted manuscript is too long although the Authors include a great amount of references. For example, a concise content in Introduction and removal of other recognitions only the fingerprint left (merged into Introduction) is suggested. Short description of the fundamental theory and importance of certain research work is enough.
Also, there are a few of unclear and inaccurate presentation in the manuscript as below.
1> Reduction of the content in Abstract.
2> Description of Disadvantage in Table 1 would be similar to that of advantage (no verb).
3> The expression of (i) and (j) in Figure 6 are not clear and appropriate. Presenting the fingerprint result like other figures would be suitable.
4> An abrupt and unexpected sentence “Capacitive fingerprint sensors are unable to detect artificial fingerprints effectively.” at the end of the second paragraph on page 16 is unacceptable.
5> What is FPS (full name)?
6> The cost-effective sensor 837 substrate delivers a resolution of up to 50 m over a 25 × 25 mm2 area. 50 m??
7> Clean Table 6.
Further polishing writings is required.
Author Response
Dear Editor
On behalf of all the contributing authors, I would like to express our sincere appreciation for your and reviewer’s constructive comments concerning our article entitled “A review of fingerprint sensor: mechanism, characteristics, and applications”. We have carefully read and responded to each comment. In light of these comments and advice, we have made careful modifications to the manuscript.
For this manuscript:
- We have simplified the abstract.
- References were adjusted according to the manuscript.
- We have modified the word and grammar errors.
We have made significant revisions to the manuscript in accordance with the comments. The changes have been marked and explained in the revised manuscript. Hope you find this revised version satisfactory. If you need to revise it again, please contact us and we will make the corresponding revision in time.

Reviewer 2 Report
Dear Authors,
The proposed paper entitled “A review of fingerprint sensor: mechanism, characteristics, and applications” is a review of sensors dedicated to identification based on biometric and especially on fingerprint. Three major physical principles are investigated: optical, capacitive and ultrasonic. Architectures, characteristics and applications are described for each physical principle involved.
The overall structure of the article is good and the bibliography is extensive, giving the article a relevant overall impression.
However, there are a number of aspects that need to be improved to perfect this article, knowing that a review is a complex exercise.
In addition, there are too many imperfections in the presentation that detract from the clarity of the text, which is why I am still asking for major corrections.
Here are my specific comments, which I hope will help the authors to improve their article.
- First of all, in terms of form, the English language needs to be revised and there are a number of typographical errors. Here are just a few examples, but there are many more:
o p.3, l.92-93: “difficult to disappear”: what does it mean?
o p.3-4, l.111-112: “… the human iris may attain perfection between …”: Is the language correct? I do not understand what the authors mean …
o p.3, l.100: “… is social life …” à “… in social life…”
o Table 1: “Ficial” à “Facial”
o Fig.2 p.6: “tirades” à “triades” ??? or something else ???
- Still in a general context, it is generally accepted to quote the reference immediately after the name(s) of the author(s) and not at the end of the sentence.
- About the form, some paragraphs are in “justified” mode, others are not, so the whole thing needs to be standardised.
- In the introduction, there is a mistake in the presentation of the section 3 that does not concern optical but capacitive sensors: cf. p.2, l.76.
- Figure 1 is clearly misplaced. Moreover, it is not commented in the text. For example, this one would be useful to introduce the problematic of the fingerprint recognition in section 2.5.
On the other hand, Figure 1 has to be reworked because many terms are not suitable:
o “Mini-equipment” à “Compacity”
o “Good cost” à “Low cost”
o “Machinism” à “Physics” or “Physical principle”
o As it is the first time that acronyms TFT, MEMS and OCT are written, they need to be explained.
- Tables as Table 1 can be very interesting and relevant but in the present case, information is quite poor, not very clear and moreover not referenced, hence unsubstantiated characteristics.
For example, what is the meaning of “Collect miniaturization”?
“The identification quality needs to be improved” is not a disadvantage … moreover, the proximity of fingerprint of relatives could be a disadvantage or the accessibility of fingerprint when touching an object … thus, leading to security problems.
About security problems, could the authors comment the comparison between “finger vein” and “fingerprint” anti-fake characteristic.
What data are classifications “High”, “Medium”, and “Low” based on?
- p.5, l.192: The authors write: “Figure 3 depicts the three primary categories …” but it is not the case. Either there is a mistake or a figure is missing.
- Figure 5 is referenced before Figure 4.
- Qualitative expressions as “was low”, “smaller in size” should be avoided in a review or they should be supported immediately by a quantification of the characteristics in question.
- Tables 5 and 6 (and to a lesser extent 3,4 and 7) need to reworked. The format chosen means that most of the headings are truncated, making the tables very difficult to read. The information needs to be selected more precisely, and readers can refer to bibliography if they wish to find out more.
It is regrettable that the general characteristics of Table 1 in terms of performance, cost and anti-fake are not clearly repeated and underlined throughout the article for the various devices presented. That is what readers can expect from a review.
Best regards.
Dear Authors,
See the general comments ...
Best regards.
Author Response

(The authors gave the same response as above.)

Reviewer 3 Report
This work presents several fingerprint acquisition techniques and analyzes acquisition types and structures,and the pros and drawbacks of various sensor type are discussed. It is a very interesting review paper for the application of IOT. I recommend its publication as it is.
Author Response

(The authors gave the same response as above.)

Round 2
Reviewer 2 Report
Dear Authors,
First of all, I would like to thank the authors for the time and effort they put into answering the reviewers' many questions and improving their paper.
In my opinion, the changes make the paper acceptable for publication.
However, to improve the presentation of the article, there are still a few small formal points that would undoubtedly be worth correcting. But as these are not essential, I leave to Editor to decide whether or not to require the authors to do so before publication (As several points are exclusively related to the final formatting of the article).
Here are these formal points :
- Personally, I'm not a fan of word cuts and I find some of them irrelevant: for example, “op-eration”, “mutu-al capacitance”.
- As already pointed out in my first report, the text is sometimes in justified mode and sometimes not. It would be clearer if all the text were in justified mode for example.
- Figure 1 is still not referenced in the text.
- When a reference concerns several authors, indicate the first (which has been done) and “et al …”. For example, p. 33, “Malgheet et al [33] …”.
- I don’t know what the policy of “Micromachines” is on this subject, but what about publishing a figure taken in its entirely from a previous article? For example, Figure 2n is taken from reference [13]. Should permission be sought from the authors, and has this been done?
- Lastly, Figure 6 and its title are not on the same page. Pay attention to the final format …
Best regards.
